# Lynch Syndrome: From Multidisciplinary Management to Precision Prevention

**DOI:** 10.3390/cancers16050849

**Published:** 2024-02-20

**Authors:** Arianna Dal Buono, Alberto Puccini, Gianluca Franchellucci, Marco Airoldi, Michela Bartolini, Paolo Bianchi, Armando Santoro, Alessandro Repici, Cesare Hassan

**Affiliations:** 1Department of Gastroenterology, IRCCS Humanitas Research Hospital, Via Manzoni 56, 20089 Rozzano, Milan, Italy; gianluce.franchellucci@humanitas.it (G.F.); marco.airoldi@humanitas.it (M.A.); michela.bartolini@humanitas.it (M.B.); alessandro.repici@hunimed.eu (A.R.); 2Medical Oncology and Haematology Unit, Humanitas Cancer Center, IRCCS Humanitas Research Hospital, Via Manzoni 56, 20089 Rozzano, Milan, Italy; alberto.puccini@humanitas.it (A.P.); armando.santoro@hunimed.eu (A.S.); 3Department of Biomedical Sciences, Humanitas University, 20072 Pieve Emanuele, Milan, Italy; 4Clinical Analysis Laboratory, Oncological Molecular Genetics Section, IRCCS Humanitas Research Hospital, 20089 Rozzano, Milan, Italy; paolo.bianchi@humanitasresearch.it

**Keywords:** Lynch syndrome, mismatch repair, immunotherapy, endoscopic surveillance

## Abstract

**Simple Summary:**

Lynch syndrome (LS) stands as the predominant inherited cancer condition at present. Currently, surveillance is based on genotype-driven strategies and mostly comprises both invasive and non-invasive medical assessments for early diagnosis. Effective preventive strategies for interrupting the biological sequence of cancer development are yet to be established. Recent findings from randomized controlled trials indicate potential preventive functions of resistant starch and/or aspirin in Lynch syndrome. The use of immunogenic frameshift peptides for vaccination appears to be a promising approach for both treating and preventing Lynch syndrome-associated cancers, based on pre-clinical and early phase 1/2a studies.

**Abstract:**

Background and Aims: Lynch syndrome (LS) is currently one of the most prevalent hereditary cancer conditions, accounting for 3% of all colorectal cancers and for up to 15% of those with DNA mismatch repair (MMR) deficiency, and it was one of the first historically identified. The understanding of the molecular carcinogenesis of LS tumors has progressed significantly in recent years. We aim to review the most recent advances in LS research and explore genotype-based approaches in surveillance, personalized cancer prevention, and treatment strategies. Methods: PubMed was searched to identify relevant studies, conducted up to December 2023, investigating molecular carcinogenesis in LS, surveillance strategies, cancer prevention, and treatment in LS tumors. Results: Multigene panel sequencing is becoming the benchmark in the diagnosis of LS, allowing for the detection of a pathogenic constitutional variant in one of the MMR genes. Emerging data from randomized controlled trials suggest possible preventive roles of resistant starch and/or aspirin in LS. Vaccination with immunogenic frameshift peptides appears to be a promising approach for both the treatment and prevention of LS-associated cancers, as evidenced by pre-clinical and preliminary phase 1/2a studies. Conclusions: Although robust diagnostic algorithms, including prompt testing of tumor tissue for MMR defects and referral for genetic counselling, currently exist for suspected LS in CRC patients, the indications for LS screening in cancer-free individuals still need to be refined and standardized. Investigation into additional genetic and non-genetic factors that may explain residual rates of interval cancers, even in properly screened populations, would allow for more tailored preventive strategies.

## 1. Introduction

Lynch syndrome (LS) is an inherited, cancer-prone, autosomal dominant disorder caused by impaired DNA mismatch repair (MMR), which proofreads genome integrity after DNA replication [1]. Germline pathogenic variants in any of the four MMR genes (*MLH1*, *MSH2*, *MLH6*, and *PMS2*) result in the loss of function of the corresponding protein [2]. LS-associated tumors exhibit microsatellite instability (MSI) (i.e., modifications in the lengths of tandem repeats within microsatellite regions) as distinctive molecular phenotypes [1,2]. The first genetic loci associated with LS were mapped back in the 1990s. Today, it is estimated that approximately 1 in every 300 individuals in North America may carry pathogenic variants in an MMR gene [3], and LS accounts for 3% and 6% of all colorectal cancers (CRCs) and endometrial cancers, respectively [4,5]. In addition to CRC and endometrial cancer, individuals with LS can develop multiple primary tumors, including cancers of the stomach, ovaries, urinary tract, and pancreas [6]. In the last decade, DNA mismatch repair (dMMR)-deficient tumors have been recognized to have both favorable prognoses, with high survival rates, and intrinsic susceptibility to immunotherapy (i.e., anti-PD1/PD-L1), findings that have transformed modern oncology. 

In this review, we aim to summarize the current evidence and recent molecular discoveries in LS, and to discuss their implications for genotype-driven strategies for surveillance, personalized cancer prevention, and treatment.

## 2. Materials and Methods

For this narrative review, PubMed was searched for relevant studies conducted up to December 2023, using the following terms: “Lynch syndrome”, “mismatch repair”, “hereditary colorectal cancer”, “*MLH1*”, “*MSH2*”, “*MSH6*”, “*PMS2*”, “*EPCAM*” and “carcinogenesis” and/or “surveillance”, and/or “prevention” to identify pertinent publications exploring the molecular pathogeneses of CRC in in patients with Lynch syndrome, as well as the related therapeutic and preventive applications. Both animal and human studies were included.

## 3. Molecular Genetics and Phenotypic Heterogeneity

Patients with LS are carriers of constitutional pathogenic mutations, resulting from the deficiency of MMR genes (dMMR), such as *MLH1*, *MSH2*, *MSH6*, and *PMS2*, or *EPCAM* deletion (which causes the epigenetic silencing of *MSH2*) [7,8].

LS is an autosomal dominant disorder that increases the lifetime risk of developing several malignancies, the most common of which is CRC [9]. Molecular alterations are not only crucial for diagnosing the syndrome, but can also serve as clinical guides for managing LS, based on the different clinical phenotypes caused by each mutated gene [9,10,11]. LS is caused by an alteration in the mismatch repair (MMR) mechanism [12], due to germline mutations that result in a loss of function of MMR; a 1000-fold increase in the error rate during DNA synthesis is observed in individuals with LS [13]. The loss of function is mostly due to nonsense or frameshift changes in LS-predisposing MMR genes, although missense changes, resulting in single amino acid substitutions, are also notable, accounting for 30–60% of all LS mutations [2]. Germline mutations in LS have been extensively studied, with *MLH1* and *MSH2* accounting for approximately 70% of all mutations, followed by *MSH6* and *PMS2* mutations [7]. *MSH2* is located on chromosome 2 and was first mapped and cloned in 1993, due to a single-nucleotide mutation, specifically a T-C substitution in a splice acceptor site [10]. In addition to classic germline mutations in *MLH1*, *MSH2*, *MSH6*, *PMS2*, and *EPCAM*, constitutional epimutations have been described in both MSH2 and MLH1 to also cause LS [14]. Epimutations include modifications, such as DNA methylation or histone modification, which can affect gene expression without altering the underlying DNA sequence. Specifically, two different studies have reported the absence of *MSH2* and *MSH6* expression in families with LS, but without germline changes in gene sequences [15,16]. The first study investigated a family with a history of hereditary non-polyposis colorectal cancer (HPNNC), but without any defined molecular alterations [15]. In this sample, a single-nucleotide alteration was found, resulting in the presence of methylation in the *MSH2* promoter [15]. Lindsberg et al. reported the phenomenon of *MSH2* silencing by transcriptional read-through of a neighboring gene in the sense direction. Deletions of one of the terminal exons of the *EPCAM* gene, located just upstream of *MSH2*, were observed in two distinct German and Chinese families [16]. These genetic changes result in a loss of the transcription termination signal of *EPCAM*, which is continued in the *MSH2* genes, resulting in non-functional transcription [16]. Germline epimutations of *MLH1* in LS were first described in 2001 [17]. *MSH2* and *MLH1* epimutations have been observed in various somatic tissues, with high degrees of mosaicism, approaching 10%. These single-allele mutations are associated with other epigenetic mutations in colonic, rectal, and endometrial cells, leading to malignant transformations. The molecular phenotype differs significantly from sporadic *MLH1* epigenetic mutations, where biallelic methylation is widely diffused only in CRC cells [18]. Lately, several single-nucleotide polymorphisms in *MLH1* (i.e., the c.-27C>A, c.85G>T, and c.544A>G variants), as well as in the *MSH2* (c.2063T>G in exon 13), have been identified as being responsible for LS [19,20,21]. 

To enhance understanding of colorectal cancer mutations, the Human Variome Project and the International Society for Gastrointestinal Hereditary Tumours (InSiGHT) have established an international database in which to store and classify all inherited variations of colorectal cancer genes [22]. This project proposes a five-level classification for MMR gene variations, as follows: class 5 (pathogenic), class 4 (probably pathogenic), class 3 (unknown), class 2 (probably not pathogenic), and class 1 (not pathogenic) [22].

As previously discussed, different genetic alterations within LS correspond to different clinical phenotypes and rates of CRC and other malignancies. Overall, patients with LS have a cumulative risk of CRC of 15–70% by the age of 70 years [23]. Mutations in *MLH1* and *MSH2* are more likely to cause classic LS, as first described in 1967 [24], and fulfil the Amsterdam I criteria [25]. The literature reports a wide range of lifetime risks of developing colorectal cancer (CRC) in patients with *MLH1* and *MSH2* mutations, ranging from 30% to 97% [26,27,28,29,30,31] in male patients, as described by Stoffel and colleagues [32].

In LS, the incidence of CRC is strongly associated with male sex, especially for *MLH1* mutations [33]. *MLH1* and *MSH2* mutations also result in an earlier age of cancer diagnosis compared to *MSH6* and *PMS2* mutations, as the diagnosis of CRC occurs at around 27–42 years of age for *MSH2/MLH1* mutations [34,35], whereas for *MSH6* and *PMS2* mutations, CRC occurs at 54–63 and 47–66 years of age, respectively [9,36].

Worldwide, the onset of CRC occurs 10 years later than in patients with *MLH1/MSH2* mutations. This was confirmed in the prospective research by Moller et al. [37], wherein no cases of CRC were diagnosed before the age of 40 in patients with *MSH6* or *PMS2* mutations [38]. Overall, *MSH6* mutations, in contrast to *MLH1* and *MSH2* mutations, appear to confer a low risk of developing CRC, of around 20% [37]. *PMS2* is thought to be the MMR mutated gene that confers the lowest risk of developing CRC, which is estimated to be around 10% by the age of 70 [37], consistent with an overall lower penetrance of mutations occurring in this gene [36]. However, LS is known to increase the risk not only of CRC, but also of other oncological diseases [39]. After CRC, the second most common neoplasia associated with LS is endometrial cancer, with a lifetime risk ranging from 21% to 60% [9]. *MSH2* mutations are most strongly associated with endometrial cancer [29,40], with a cumulative risk at 70 years of 54% [9,41]. In terms of extra-colon cancer risk, germline mutations in *MSH6* and *PMS2* have been associated with increased risks of both breast cancer and ovarian cancer, with a doubling of the likelihood of developing breast cancer by the age of 60 years [42]. Finally, specific *MLH1* pathogenic variants (i.e., c.2181_2182delCA and c.229T>C) have been associated with certain extra-colonic cancers in LS, particularly with pancreatic cancer [43]. Table 1 elucidates the absolute risks of developing the main types of cancers associated with LS. 

## 4. CRC Development in Individuals with LS

dMMR alterations result in a loss of the ability to repair DNA mismatches, promoting accelerated carcinogenesis [24,39]. However, phenotypic penetrance does not reach 100% of cases, and, therefore, not all patients with LS will develop cancer during their lifetime. In addition, significant variability in age and type of tumor at presentation has been described [2]. The exact mechanism underlying carcinogenesis in patients with LS is still not fully understood, although many aspects have been elucidated over the years. 

Transgenic dMMR mice, with proven heterozygosity for *MLH1* and *MSH2*, showed reduced expression of MMR proteins and increased levels of genomic frameshift deletions, but this is insufficient to explain the whole process of tumorigenesis [44]. Even in somatic cells with a heterozygous genotype for a specific MMR gene, a second alteration must occur to eliminate the wild-type allele of the MMR gene, according to Knudson’s two-hit hypothesis [45].

In addition, different levels of gene expression may be required, depending on the function assigned to the different MMR proteins, highlighting the variable importance of their roles in DNA integrity. For instance, higher levels of *MLH1* are required for DNA damage signaling and arrest at the G2/M cell cycle checkpoint during replication, compared to DNA MMR [46]. On the other hand, there are several pathways that may lead to a dMMR predisposition, such as the decreased messenger RNA expression of *MLH1* and the aberrant chromosome segregation genes observed in the normal colonic mucosa of *MLH1þ*/– and *MLH1þ*/þ mice [47]. Interestingly, dMMR niches have been described in the normal colonic mucosa [48] and the normal endometria [49,50] of dMMR carriers, whose true role in carcinogenesis remains to be clarified. Lee et al. demonstrated the genomic stability of histologically normal epithelial cells from LS patients through whole-genome sequencing of their tissues [51]. However, they identified a dMMR crypt, with an increased mutational burden and a dMMR-related signature, that may represent a very early stage of CRC [51]. Several pathogenetic theories have been proposed to explain the rate of interval cancers and the failure of endoscopic excision of prior adenomas to prevent cancer [52]. One of the main hypotheses is that cancers arising from dMMR crypts may develop as endoscopically undetectable lesions, in contrast to the well-known histological progression from adenoma to adenocarcinoma [52]. Furthermore, immunohistochemical analysis showed that patients with LS appeared to have MMR defects, particularly in polyps larger than 8 mm, whereas this was much less common in polyps smaller than 8 mm [53]. This may suggest that polyp development may not have occurred prior to the somatic loss of function of the wild-type allele, but may only manifest after this catastrophic event.

Pathogenic *MLH1* variants are among the lesions that often escape endoscopic surveillance, whereas adenomas are more frequently identified in *MSH2* carriers [54]. Somatic *APC* variants are less common in *MLH1*- than in *MSH2*-deficient tumors, whereas C catenin beta 1 (*CTNNB1*) variants are more common in *MLH1*-deficient tumors [54]. Differences in the developmental pathways of screen-detected and non-screen-detected CRCs have also been reported.

One explanation for this discrepancy may be that KRAS codons 12 or 13 are rarely detected in tumors identified during routine endoscopic surveillance, and adenocarcinoma sequence alterations have evolved from MMR deficiency [55]. *CTNNB1* variants were initially hypothesized to be involved in non-polyposis tumorigenesis [56]. In a study comparing *PMS2*- and *MLH1*-deficient CRC, no somatic *CTNNB1* variants were detected in *PMS2*-associated CRC (0/20, 0%), whereas a significant number of *CTNNB1* variants were present in *MLH1*-associated CRC (14/24, 58%) [57]. Furthermore, *PMS2* deficiency in adenomas is preceded by *KRAS* alterations, as is CRC in *PMS2* variant carriers [57]. This is consistent with the absence of *PMS2*-related early-onset CRC, detected by surveillance colonoscopy, also suggesting that the dMMR crypt foci do not contribute to *PMS2*-deficient tumor development [58]. This seems to fit well with the lack of *PMS2*-related early-onset CRC observed during surveillance colonoscopy. It also suggests that *PMS2*-deficient carcinogenesis is not affected by the dMMR crypt foci pathway [58]. Of note, it is widely known that LS-associated cancers are highly immunogenic (“hot tumors”), and that they are more frequently treated with immunotherapy, compared to other tumors characterized by lower tumor cell density (“cold tumors”) [59]. In fact, their hypermutated genomic profile leads to an abundance of effector and memory T-cells in the tumor microenvironment and in the tumor core, as well as at the invading front. A mutational burden above ten changes per mega base is typically present in MSI-H tumors [2].

In terms of potential protective factors, acetylsalicylate has been shown to be effective in the primary and secondary prevention of CRC, although its modulatory role on the immune milieu of normal bowel mucosa remains unclear. 

A double-masked, randomized clinical trial showed that preventive administration of daily 600 mg of acetylsalicylate daily for up to four years reduced the incidence of CRC by half, with an average follow-up of more than ten years [60]. Interestingly, naproxen has also been studied in this setting. When administered at 220 mg or 440 mg daily for six months, this drug showed significant molecular changes at the level of the intestinal mucosa in a mouse model. In particular, naproxen promoted the activation of various immune cell types, the reduction in prostaglandin E2 levels, the decreased synthesis of stem cell markers, and modulated epithelial differentiation markers [61].

Finally, different immune profiles are present in the normal intestinal mucosa of individuals with LS, both with and without cancer. CD3-, FOXP3-, and CD8-positive T-cell densities were higher in dMMR variant carriers without CRC, compared to non-LS individuals and patients with LS affected by CRC [62]. As cancer becomes diagnostically detectable, the densities of normal intestinal immune cells appear to be significantly reduced, compared to those in early stages. Consequently, we can cautiously suggest that the epithelial immune microenvironment may be a modifiable risk factor, as suggested by the known success of NSAIDs and immune checkpoint inhibitors in the treatment of MSI-H CRC. Given this evidence, it could be considered, in the near future, as a risk stratification biomarker and play an active role in cancer surveillance. The development of microsatellite instability is often seen at the same genomic locations in MSI cancers (regardless of the original tissue type), supporting evidence for a common DNA frameshift that triggers breakpoints in the translation process, even when the resulting peptides are related to different tumor types. The above mechanism facilitates counter-selection of cell clones by highly immunogenic frameshift peptides [63]. The most common frameshift mutations in MSI tumors open the gates for immunotherapeutic efficacy and immune pre-prevention by using the resulting peptides as neoantigens for adaptive immunity [64,65].

## 5. Genetic Counselling 

As CRC surveillance has been shown to reduce CRC risk and mortality in LS [66,67,68] the diagnosis of LS represents a fundamental moment in its management. The most important question in a clinician’s clinical routine is to understand who should undergo genetic counselling for the diagnosis of LS [9]. Genetic testing for LS has three distinct aims: to confirm the diagnosis of LS in a patient and or their family, to determine the status of family members at risk, and to define the management of affected and unaffected individuals. Although LS is the most common cause of hereditary CRC, it still suffers from a significantly high rate of underdiagnosis [69]. The classical Amsterdam [70] and Bethesda [71] clinical criteria for defining patients at high risk of LS lack specificity and sensitivity. This problem has been overcome by more recent tools, such as various clinical prediction models and the universal screening strategy. The classical criteria used in clinical practice have been shown to be even less accurate than in clinical trials, because they require a careful analysis of the patient’s past medical and family history, which is sometimes not possible in clinical practice. Additionally, they have a sensitivity of less than 50% in diagnosing patients with *MSH6* and *PMS2* mutations [72]. The clinical prediction models are based on computational methods that include various parameters such as age, gender, family history of CRC or endometrial cancer, tumor location, and molecular status. The principal algorithms developed include the following: MMRpredict, MMRpro, and the PREMM1, 2, and 6 models. Each of them has shown better performance than the classical clinical criteria [12,73,74,75]. Genetic counselling is recommended for patients with a 5% risk of LS, as assessed by MMRpro and the PREMM1, 2, 6 models [76]. This approach could also be cost effective in the genetic screening for LS [77]. To reduce the diagnostic gap in LS, a universal screening strategy has been increasingly adopted, which consists of testing for specific molecular features in all CRCs and endometrial cancers in order to detect the specific feature of CRC and endometrial cancer associated with LS [78]. This strategy helps to reduce the gap in the diagnosis of LS. If LS features are detected, genetic counselling should be offered to at-risk family members, starting with first-degree relatives [79,80,81]. Various tests are available to confirm LS with good performance. These include immunohistochemistry and microsatellite instability testing, as well as multigene testing through next-generation sequencing [82,83]. In fact, multigene testing is widely available and may represent a valid alternative to the standard test for traditional disease-specific mutations, potentially allowing for the detection of germline mutations causing known CRC, even when unrelated to LS [4,66,84]. Multigene testing should always be performed after obtaining informed consent from trained physicians [85,86]. Analysis of tumor tissue is the preferred choice for the diagnosis of LS in order to search for the same mutations in the proband’s siblings. However, if tumor tissue is not available, or if a universal screening strategy has not been adopted, patients with a probability of more than 5%, using clinical prediction models, or who fulfil classic clinical criteria should undergo genetic counselling [9,87].

## 6. Surveillance Recommendations

Endoscopy remains the gold standard method for colorectal cancer (CRC) surveillance in patients with LS [23,66]. The surveillance program is tailored to the cumulative risk of CRC development in LS patients (Table 2). Based on the described phenotype of CRC in LS patients, alterations in *MLH1* and *MSH2* are associated with higher incidences of CRC, as well as earlier onset, compared to alterations in *MSH6* and *PMS2* [37,38]. Various endoscopic societies recommend starting endoscopic CRC surveillance at 25 years of age for patients with *MLH1* and *MSH2* mutations (Table 2) [23,66]. However, there is disagreement among these societies regarding the CRC surveillance program for patients with *MSH6* and *PMS2* alterations. Recently published European and British guidelines recommend starting screening in these patients at the age of 35 years [37], which differs from the guidelines in America and Japan. These guidelines were published before the publications of the of the prospective evidence obtained by Finnish research, and they also marked the start of CRC surveillance for patients with *MSH6* and *PMS2* mutations at 25 years of age [88].

The recommended surveillance interval for colonoscopy has now been set at 2 years by the European, Japanese, and American endoscopy societies [23,66,89]. Two well-conducted prospective studies have clarified the optimal timing of endoscopic surveillance. These studies were based on prospective registries of German, Finnish, and Dutch patients, including more than 2000 patients, with a cumulative observation time of 23,309 person-years. The cumulative risk of CRC ranged from 4.8% to 18.5%, depending on whether the patients were classified as low or high risk. The study prospectively assessed the risk of CRC and it analyzed the risk, as stratified by the three different surveillance methods used in the participating countries. No reduction in CRC risk or tumor stage was observed when the German annual surveillance schedule was compared with the 1/2-year or 2/3-year schedule of Dutch and Finnish patients [90]. The study evaluated the risks of developing adenomas, advanced adenomas, and CRCs in a cohort of the same patients who underwent at least two surveillance colonoscopies over a 10-year period. The results showed that *MSH2* patients had a significantly higher risk of developing adenomas compared to *MSH6* and *MLH1* patients. The risk of advanced adenoma was similar in *MSH6* and *MLH1* patients, at 7.7% (95% CI 6.0–9.4%) and 9.4% (95% CI 5.4–13.4%), respectively, and significantly lower in MSH2 patients, at 17.8% (95% CI 14.6–21%). Ten years after the index colonoscopy, the risk of colorectal cancer (CRC) was similar in *MLH1* and *MSH2* patients, at 11.3% (CI 9.4–13.2%) and 11.4% (CI 8.9–14.0%, p: 0.468), respectively, despite differences in the incidence of previous precancerous lesions. However, the risk of CRC was significantly lower in MSH6 carriers, at 4.7% (95% CI 1.8–7.7%; *MLH1* vs. *MSH6* p: 0.01; *MSH2* and *MLH1* p: 0.03). These results may lead to a revision of current recommendations for screening surveillance in LS. It also suggests a different pathogenic development of colorectal cancer in carriers of the three different mutations, *MSH2*, *MSH6*, and *MLH1* [54]. Once surveillance intervals have been established, it is important to determine how this surveillance program should be carried out. Recent research, involving more than 800 patients, has shown that high-quality colonoscopy is associated with a reduction in the interval of CRC and an increase in the detection of adenomas [91]. Adenoma detection was significantly influenced by complete colonoscopy, adequate bowel preparation, and chromoendoscopic evaluation. On the other hand, post-colonoscopy CRC was only significantly influenced by a colonoscopy interval of less than 3 years. High-definition colonoscopy, adequate bowel preparation, and complete colonoscopy showed reductions, but they did not reach statistical significance. Nowadays, various societies recommend high-quality endoscopic surveillance every 2 years for asymptomatic patients with LS. However, there is no preferred method between high-definition white light (HD-WL) and chromoendoscopy [23,66,92]. Recent research does not provide reliable evidence to determine which type of visualization, between high-definition white light and chromoendoscopy, is better for detecting CRC precursor lesions [23,66,88,92]. The role of chromoendoscopy in the detection of precancerous lesions in LS was emphasized as the preferred method of endoscopic surveillance at the beginning of the 21st century. However, this role has been underestimated over the past five years. The potential overestimation of the chromoendoscopy may be due to several factors, including the lack of high-definition imaging, small study sample size, and lack of back-to-back tandem studies. It is worth noting that the majority of trials were conducted with chromoendoscopy as the second surveillance method, which may have introduced an intrinsic bias [93,94,95,96]. Several randomized controlled studies have been conducted to investigate which view, HD-WL or chromoendoscopy, is better for surveillance in LS. A randomized multicenter experiment, conducted in 22 expert centers with 357 patients, showed no difference in the primary outcome of polyp detection rate (PDR) between the LCI group and the high-definition white light endoscopy group [97]. Furthermore, a study by Hamstra et al. found that LCI was ineffective for increasing the adenoma detection rate in the right colon and reducing the risk of CRC at two years [98]. A more recent multicenter study also confirmed the non-inferiority of HD-WL compared with pan-colonic chromoendoscopy for ADR in LS [99]. In recent years, AI has become increasingly involved in the detection of polyps and adenomas, as evidenced by several studies in the screening of non-LS populations [100,101,102,103]. However, only two randomized trials of AI in LS have been conducted, and preliminary data require further investigation to provide solid evidence for the use of AI in LS patients [104,105]. 

Concerning extra-intestinal cancer surveillance, while screening for endometrial and ovarian cancers is available in certain contexts, there is inadequate evidence regarding its clinical effectiveness. As a result, most guidelines advocate for risk-reducing surgical interventions, such as total hysterectomy and bilateral salpingo-oophorectomy, which are typically recommended between the ages of 35 and 40, or after completing childbearing [106].

Finally, while research indicates a prevalence, ranging from 42% to 51%, of breast cancers in women with LS, there is a lack of adequate data to conclusively establish an elevated risk of breast cancer in LS. Consequently, decisions regarding breast cancer risk management should be grounded in individual and family medical histories [106]. 

Table 2 elucidates cancer surveillance strategies, based on genotype, in LS patients, according to NCCN [106] and European guidelines [23,84].

## 7. Treatment of LS Cancers; Immunotherapy and Immunoprevention

Tumors associated with LS exhibit heightened burdens of insertion and/or deletion mutations within microsatellite DNA regions. This occurs due to the persistence of DNA replication errors, leading to the accumulation of a substantial number of frameshift mutations encoding immunogenic proteins in the tumors of affected individuals [107]. Additionally, specific characteristics, like higher tumor-infiltrating T-Lymphocytes (TILs), as well as overexpression of immune checkpoint receptors and ligands, mainly PD-1 and PD-L1, which suggest a local inflammatory response, have been reported [107]. Additional findings emphasize the relevance of the immune surveillance process in LS. Even before the diagnosis of cancer, frameshift peptides (FSPs) can induce both humoral- and cellular-mediated immune responses, resulting in the detection of antibodies induced by FSPs and effector T-cell responses specific to FSPs in LS patients. Interestingly, T-cells reactive to FSPs are also found in LS patients who do not have CRC, but they are absent in individuals without LS and in CRC patients whose cancers do not have MMR deficiency [108]. As cancer develops, immune surveillance mechanisms break down, and the presence of leukocytes infiltrating the tumor may indicate an ineffective attempt to initiate an anti-tumor immune response [109]. These findings provide a basis for immune-modulatory therapies, such as immune checkpoint inhibitors (ICIs), which can reactivate an exhausted immune response [110]. dMMR tumors showed the most impressive response to immune checkpoint inhibitors, independent of tumor type. This led to the first tumor-agnostic approval of ICIs in patients with metastatic disease [110,111,112]. In 2017, anti-PD-1 pembrolizumab was approved by the U.S. Food and Drug Administration for both adult and pediatric patients with unresectable or metastatic dMMR/MSI-H solid tumors. The approval was granted based on data from the phase III KEYNOTE-177 trial, which enrolled 149 patients with dMMR/MSI-H. The study showed superiority for first-line pembrolizumab over chemotherapy, with a median progression-free survival (PFS) of 16.5 months, compared to 8.2 months (hazard ratio [HR] = 0.60, 95% CI, 0.45–0.80, *p* = 0.0002) [113,114]. The KEYNOTE-177 study did not report a statistically significant overall survival (OS) benefit with pembrolizumab vs. chemotherapy, likely due to a high rate (60%) of crossover to ICIs in the chemotherapy arm after progression.

Several phase II trials in the context of CRC have shown that anti-programmed cell death-1 (PD-1), either alone or in combination with anti-cytotoxic T-lymphocyte-associated antigen-4 (CTLA-4), demonstrates high efficacy and confers a survival advantage in MSI-H/dMMR mCRC [115,116]. Recently, it has been observed that early stage dMMR cancers exhibit an even greater response to ICIs when compared to metastatic disease. Promising results, in terms of pathological complete response, were reported in the NICHE-2 study, with a single dose of ipilimumab and two doses of nivolumab, followed by surgery. In locally advanced rectal cancer, the administration of neoadjuvant dostarlimab led to a persistent complete response in a “watch and wait” and organ-sparing approach [59,117]. Moreover, the NICHE-2 trial recently highlighted that preoperative ICIs were correlated with an increased pathological complete response rate in patients with LS-associated colon cancer, compared to sporadic MSI-H primary colon cancer [117]. Consistent with these findings, a retrospective study, carried out by Colle et al. (2023), of patients treated with ICIs for mCRC with a strict definition of LS, suggests that LS is protective against PFS events. This could be attributed to various clinical, histological, and immunological characteristics. Notably, LS-associated tumors exhibit higher T-cell infiltration, a greater mutational burden, and an increased presence of neoantigens, compared to patients with sporadic dMMR. These differences may account for the discrepancy in prognosis observed in these two populations [118]. This evidence has fueled interest in the potential use of ICIs for cancer prevention in individuals with LS. Following the results of the study conducted by Heudel et al. (2021), Cercek et al. (2023) reported data from 172 cancer-affected patients with LS, attempting to quantify the persistent risk of pre-malignant and malignant neoplasia in patients previously treated with ICIs. As reported in the study, 12% of these patients developed subsequent malignancies, and 39% were found to have pre-malignant colonic polyps. Furthermore, there was no difference in the overall rates of tumor development within matched pre- and post-ICI follow-up periods. This study suggests that exposure to ICIs does not permanently eliminate the risk of either pre-malignant or subsequent malignant neoplasms in patients with LS [119,120]. 

Successful outcomes with immunotherapy using ICIs have unequivocally demonstrated the immune system’s capability to generate potent antitumor responses, resulting, in some cases, in complete response. These findings have paved the way for the development of cancer vaccines in both prevention and interception settings. Indeed, in these scenarios, local immunity within the tumor molecular environment (TME) is less compromised, and there remains low clonal heterogeneity of tumor antigens [121]. A clinical phase I/II trial was conducted in patients with history of or current dMMR CRC to evaluate the safety and immunogenicity of an FSP-based vaccine, utilizing FSP neoantigens derived from mutant AIM2, HT001, and TAF1B. Although no safety concerns were reported, all patients exhibited consistent induction of both humoral and cellular immune responses. Furthermore, a heavily pretreated patient with bulky metastases maintained stable disease and stable CEA levels for a period of 7 months [122]. In addition to the peptide-based FSP vaccination approach, ongoing studies are exploring alternative vaccination strategies. These include viral vector-based FSP antigen delivery, which offers a significantly larger number of antigens (over 200). Nouscom S.R.L. recently developed a viral-vectored vaccine called Nous-209, which encodes 209 shared FSPs targeting MSI tumors. The phase Ib/II clinical trial of Nous-209 is currently underway, assessing its efficacy in eliciting recurrent neoantigen immunogenicity and cancer immune interception in patients with LS. The study aims to evaluate the safety of Nous-209 administration in LS patients and investigate its potential impact on the development of colon polyps or tumors (NCT05078866).

Incorporating immunomodulatory agents into cancer vaccines may be a hypothesis to stimulate stronger adaptive immune responses in LS patients and other high-risk populations. In this way, preclinical research has shown that vaccination with a combination of four MSI-specific FSPs enhances anti-FSP immunity in mice, whether they are naïve or have undergone *MSH2* conditional intestinal knockout. This leads to delayed tumor growth and significantly prolonged survival in vaccinated mice, compared to those who were not vaccinated. Furthermore, this benefit has been proven to be significantly enhanced in a cooperative manner by combining FSP vaccination with non-steroidal anti-inflammatory drug (NSAID) treatment [123]. Meanwhile, the phase IIB clinical trial conducted by CCARE is recruiting patients with LS to assess the safety and effectiveness of the three vaccines (Tri-Ad5), combined with a protein (N-803) that will boost the vaccines’ effects, to determine if there is any effect on the risk of developing colon and other cancers in patients with LS (NCT05419011).

In the context of cancer prevention in LS, there is considerable clinical evidence supporting the potential efficacy of aspirin. However, its application in clinical practice is hindered by the potential risk of side effects, mainly bleeding. Currently, the CAPP-2 (Cancer Prevention Program 2) is the only completed clinical trial on chemoprevention in LS patients, which aims to investigate the antineoplastic effects of aspirin (at 600 mg/day) and resistant starch in LS carriers. After 20 years of follow-up, intention-to-treat analysis for CRC demonstrated a notable 35% risk reduction (HR = 0.65, 95% CI, 0.43–0.97; *p* = 0.035), with this effect being particularly significant among overweight patients. There were no significant differences in adverse events, such as bleeding or gastrointestinal issues, between aspirin-treated subjects and those receiving the placebo [60]. Despite adverse events remaining the main obstacle in the use of aspirin for primary prevention, these results confirm its effectiveness and good tolerance. Therefore, the main challenge is to identify the optimal aspirin dose to minimize these potential side effects, without reducing its efficacy. In this regard, the CAPP-3 trial is currently underway, aiming to compare the effects of different doses of aspirin (100 mg, 300 mg, and 600 mg) in patients with LS. To date, using the available data for CRC prevention, and in line with the major international guidelines, the systematic review by Serrano et al. (2022) proposes aspirin as preventive medicine at a dose of 100 mg per day [124]. Like aspirin, other NSAIDs have demonstrated comparable impacts on reducing the risk of LS-CRC; both ibuprofen and naproxen have demonstrated reductions in the risk of CRC development among LS patients, with naproxen associated with prolonged survival in this patient cohort [61,125]. In addition to decreasing levels of pro-tumorigenic prostaglandin E2, naproxen has exhibited the ability to enhance antitumor immune responses in LS mice, through rFSP vaccination in the VcMsh2 mouse study, and to increase immune surveillance in LS patients. 

Further investigation, through preclinical and clinical research, is necessary to explore the effectiveness, safety, and actual clinical relevance of combining cancer preventive vaccines with these newer immunomodulatory agents, particularly in LS patients and high-risk populations. Additionally, informatics and genomics efforts may help identify validated immunogenic shared neoantigens for LS patients, forming a foundation for new cancer preventive vaccines. These vaccines could potentially prevent not only LS-associated CRC, but also extracolonic tumors, as certain neoantigens are likely shared across the spectrum of cancers arising from LS. The attempts of our community to improve the clinical outcomes of patients with LS shall focus on promising available strategies, among which we certainly have prompt identification of patients with LS, to avoid missed diagnosis and start specific screening programs for metachronous CRC, as well as other LS-associated cancers. Secondly, whenever clinically appropriate, high-risk surveillance of patients with LS should be continued throughout and after ICI treatment. Future prospective research should focus on designing new effective drugs to formulate novel therapeutic strategies. This is crucial to address the conspicuous fraction of patients exhibiting primary resistance or developing acquired resistance during treatment with ICIs. 

An emerging aspect worthy of clinical investigation concerns a distinct subgroup of CRC, such as tumors with homologous recombination deficiency, as the frequency of DNA damage response mutation was higher in MSI-H cancers than in MSS cancers [126]. Targeting the homologous recombination system and ICIs in this subgroup might be a potential approach that deserves further efforts [127]. Nevertheless, there is an urgent need for new predictive biomarkers of resistance to ICIs among MSI/dMMR tumors. This is essential in order to identify the optimal treatment for these patients and to develop novel therapeutic approaches for MSI/dMMR mCRC patients who have experienced prior ICI treatment failure. Finally, together with technological advances in LS tumor genomic landscape profiling and the developments of improved drug designs, including cancer vaccines and innovative immunomodulatory agents, precision cancer prevention and treatment for LS carriers is becoming increasingly attainable.

## 8. Conclusions

LS requires careful investigation for diagnosis, monitoring, prevention and treatment. Despite advances in the knowledge of the molecular mechanisms of carcinogenesis and diagnostic technologies, and the development of international recommendations, LS remains underdiagnosed [128]. In particular, recent research has shown that the establishment and implementation of standardized LS screening algorithms increases the efficiency of LS identification [129,130] (Figure 1). 

In the coming years, the integration of NGS into clinical practice is certain to improve detection rates and enable more efficient, accurate, and personalized management programs for patients diagnosed with colorectal cancer. The trend in guidelines is to progressively and rapidly expand access to testing for a significant number of CRC patients. The goal of the scientific community is to identify clinically actionable findings to guide treatment and prevent further primary cancers in other organs.

There are many gaps in knowledge regarding surveillance, particularly given the high rate of interval cancers detected (i.e., in *MLH1*-mutant LS) and the lack of evidence regarding the clinical effectiveness of cancer screening for extra-colonic LS cancers. Risk stratification and tailored surveillance currently comprise the backbone of the management of LS, with emerging data on their cost effectiveness [131,132]. LS is an example of precision medicine because its identification enables the implementation of preventive measures aimed at reducing cancer incidence and mortality. Colonoscopy screening for colorectal cancer, aspirin use, and prophylactic hysterectomy and bilateral salpingo-oophorectomy for endometrial and/or ovarian cancers have been shown to be effective in reducing cancer mortality in the LS population. Nevertheless, the impacts of environmental factors on the gene-specific lifetime risk of each cancer in the LS population require further investigations in other to improve and optimize screening intervals and modalities. In our opinion, testing for dMMR/MSI-H stands out as a prime example of a universally applicable somatic test that is critical for both therapeutic intervention and genetic counseling/testing recommendations. This is due to the well-established and obvious benefits of surveillance programs for both patients and their at-risk relatives, together with the prospect of tailoring a personalized, lifelong, gene-specific management strategy for individuals diagnosed with LS. 

Preventive pharmacological measures are expected to be introduced soon, and there are preliminary and promising developments in the uses of immunotherapy and anti-cancer vaccines in the treatment of these highly immunogenic tumors associated with LS. 

## Figures and Tables

**Figure 1 cancers-16-00849-f001:**
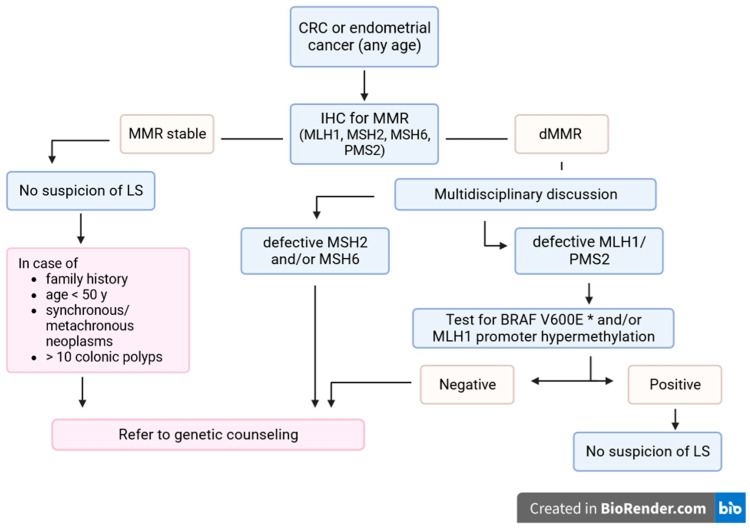
LS screening algorithm in CRC or endometrial cancer patients. * Only for CRC. MMR: mismatch repair genes; IHC: immunohistochemistry; LS: Lynch syndrome; CRC: colorectal cancer.

**Table 1 cancers-16-00849-t001:** Cumulative cancer risks in Lynch syndrome according to genotype.

Cancer Type	MLH1	MSH2/ EPCAM	MSH6	PMS2	General Population
Colorectal cancer	46–61%	33–52%	10–44%	8.7–20%	4.2%
Endometrial cancer	34–54%	21–57%	16.49%	13–26%	2.7%
Ovarian cancer	4–20%	8–28%	≤1–13%	1.3–3%	1.3%
Ureteral cancer	0.2–5%	2.2–28%	0.7–5.5%	≤1–3.7%	<1%
Gastric cancer	5–7%	0.2–9%	≤1–7.9%	Not known	0.9%
Pancreatic cancer	6.2%	0.5–1.6%	1.4–1.6%	≤1–1.6%	1.6%
Prostate cancer	4.4–13.8%	3.9–23.8%	2.5–11.6%	4.6–11.6%	11.6%

**Table 2 cancers-16-00849-t002:** Recommendations for cancer surveillance in Lynch syndrome.

Cancer Type	MLH1	MSH2	MSH6	PMS2
Colorectal cancer	Colonoscopy every 1–2 years, starting at 20–25 y	Colonoscopy every 1–2 years, starting at 20–25 y	Colonoscopy every 1–3 years, starting at 30–35 y	Colonoscopy every 1–3 years, starting at 30–35 y
Endometrial and ovarian cancers *	Pelvic ultrasound and/or endometrial biopsy every 1–2 years, starting at 30–35 y	Pelvic ultrasound and/or endometrial biopsy every 1–2 years, starting at 30–35 y	Pelvic ultrasound and/or endometrial biopsy every 1–2 years, starting at 30–35 y	Pelvic ultrasound and/or endometrial biopsy every 1–2 years, starting at 30–35 y
Ureteral cancer	Urinalysis, urine cytology, and abdominal ultrasound every 1–2 years, starting at 40–45 y	Urinalysis, urine cytology, and abdominal ultrasound every 1–2 years, starting at 40–45 y	Urinalysis, urine cytology, and abdominal ultrasound every 1–2 years, starting at 40–45 y	Urinalysis, urine cytology, and abdominal ultrasound every 1–2 years, starting at 40–45 y
Gastric and duodenal cancers	EGD every 3–5 years, starting at 30–35 y	EGD every 3–5 years, starting at 30–35 y	EGD every 3–5 years, starting at 30–35 y	EGD every 3–5 years, starting at 30–35 y

EGD: esophagogastroduodenoscopy. * Risk-reducing surgical treatment in the form of total hysterectomy and bilateral salpingo-oophorectomy should be offered from ages 35 to 40 years, or after completion of childbearing.

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
