# Peer review of "Lynch Syndrome: From Multidisciplinary Management to Precision Prevention"

_cancers, 2024, doi:10.3390/cancers16050849_

Round 1
Reviewer 1 Report
Comments and Suggestions for Authors
The authors’ review article is generally well-structured and informative. Here are some positive aspects and a few suggestions for improvement:
Positive Aspects:
Clarity and Conciseness: The revisions have improved the clarity of the text, making it easier to understand.
Precision: The sentences are more precise and focused, contributing to a clearer overall message.
Grammar and Syntax: The grammar and syntax have been improved, resulting in better readability.
Logical Flow: The logical flow of information is maintained, helping readers follow the narrative.
Suggestions for Improvement:
Transition Phrasing: Consider adding transitional phrases to enhance the flow between sentences and paragraphs. For example, in the transition from discussing preventive strategies to recent findings, you could use a phrase like "In light of these challenges, recent findings..."
Sentence Length: Some sentences are quite long and could be broken down for better readability. For instance, the sentence beginning with "The trend of guidelines..." is lengthy and could be split for clarity.
Redundancy: The phrase "True preventive strategies aiming to interrupt the biological sequence for cancer development are still lacking" could be simplified. Consider something like "Effective preventive strategies to interrupt the biological sequence of cancer development are yet to be established."
Conclusion Emphasis: In the final paragraph, consider summarizing the key points discussed in the text to reinforce the main takeaways for the reader.
Author Response
Response to the editor(s) of ‘Cancers’
Dear Editor(s),
We sincerely thank you for giving us the opportunity to consider for re-submission and potential publication a revised version of our original manuscript entitled “Lynch syndrome: from multidisciplinary management to precision prevention" by Dal Buono et al.
We kindly thank the Reviewers for the precious comments. We are pleased to know that you appreciated the topic of our manuscript. The manuscript has been significantly revised and improved according to the received suggestions. Included below you can find a point-by-point response to the remarks.
Sincerely,
Arianna Dal Buono, MD
Humanitas Research Hospital – IRCCS
Rozzano, MI, Italy
Reviewer#1
The authors’ review article is generally well-structured and informative. Here are some positive aspects and a few suggestions for improvement:
Re: Thank You for Your comment. We are glad that You appreciated our manuscript.
Positive Aspects:
Clarity and Conciseness: The revisions have improved the clarity of the text, making it easier to understand.
Re: Thank You for Your comment.
Precision: The sentences are more precise and focused, contributing to a clearer overall message.
Grammar and Syntax: The grammar and syntax have been improved, resulting in better readability.
Re: Thank You for Your comment.
Logical Flow: The logical flow of information is maintained, helping readers follow the narrative.
Re: Thank You for Your comment.
Suggestions for Improvement:
Transition Phrasing: Consider adding transitional phrases to enhance the flow between sentences and paragraphs. For example, in the transition from discussing preventive strategies to recent findings, you could use a phrase like "In light of these challenges, recent findings..."
Re: Thank You for Your comment. We improved the text as suggested.
Sentence Length: Some sentences are quite long and could be broken down for better readability. For instance, the sentence beginning with "The trend of guidelines..." is lengthy and could be split for clarity.
Re: Thank You for Your comment. We improved the text as suggested.
Redundancy: The phrase "True preventive strategies aiming to interrupt the biological sequence for cancer development are still lacking" could be simplified. Consider something like "Effective preventive strategies to interrupt the biological sequence of cancer development are yet to be established."
Re: Thank You for Your comment. We improved the text as suggested.
Conclusion Emphasis: In the final paragraph, consider summarizing the key points discussed in the text to reinforce the main takeaways for the reader.
Re: Thank You for Your comment. We improved the text as suggested.
Reviewer 2 Report
Comments and Suggestions for Authors
The authors provide a comprehensive narrative review on LS.
Since this is one of several reviews on this topic, to meka eit more appealing to the readers the authors should dedicate a separate section to inform us about any recent updates on genes or mutations or SNPs re: LS, any additional progress on genetics/genomics pertaining to the diagnosis and management of LS as well as any novel associations with additional cancers or syndromes. This addition would strengthen the significance of this review and make it unique.
Author Response
Response to the editor(s) of ‘Cancers’
Dear Editor(s),
We sincerely thank you for giving us the opportunity to consider for re-submission and potential publication a revised version of our original manuscript entitled “Lynch syndrome: from multidisciplinary management to precision prevention" by Dal Buono et al.
We kindly thank the Reviewers for the precious comments. We are pleased to know that you appreciated the topic of our manuscript. The manuscript has been significantly revised and improved according to the received suggestions. Included below you can find a point-by-point response to the remarks.
Sincerely,
Arianna Dal Buono, MD
Humanitas Research Hospital – IRCCS
Rozzano, MI, Italy
Reviewer#2
The authors provide a comprehensive narrative review on LS. Since this is one of several reviews on this topic, to make it more appealing to the readers the authors should dedicate a separate section to inform us about any recent updates on genes or mutations or SNPs re: LS, any additional progress on genetics/genomics pertaining to the diagnosis and management of LS as well as any novel associations with additional cancers or syndromes. This addition would strengthen the significance of this review and make it unique.
Re: Thank You for Your comment. We improved the text as suggested (see the paragraph molecular genetics and phenotypic heterogeneity).
Reviewer 3 Report
Comments and Suggestions for Authors
The review Lynch syndrome: from multidisciplinary management to precision prevention
by Arianna Dal Buono et al is an attempt to summarize and discuss “the genotype-driven strategies for surveillance, personalized cancer prevention and treatment” in case of Lynch syndrome (LS) cancers.
Please find below my comments and suggestions
Abstract “Background and aims: Lynch syndrome (LS) is currently the most prevalent hereditary cancer condition” - what about hereditary breast and ovarian cancer syndrome? Please provide references regarding prevalence of the most common hereditary cancer conditions.
“Significant advances have recently improved the understanding of the molecular carcinogenesis of LS tumors “ - what advances? The sentence is not informative.
“We aim to review the latest progresses in LS and discuss their implications for genotype-driven strategies for surveillance, personalized cancer prevention and treatment.” what implications? This sentence should be re-written too.
Line 49 - “DNA mismatch repair (dMMR)”, but “impaired DNA mismatch repair (MMR)” line 27. The abbreviation is inconsistent. Does the latter abbreviation means “deficient DNA mismatch repair (dMMR)?
Line 50. “Germline pathogenic variants <> results” - perhaps “result”?
Line 52. “microsatellite instability (MSI), otherwise modifications in the length of tandem repeats within microsatellite regions” - I suggest replacing this sentence with “microsatellite instability (MSI), i.e. modifications in the length of tandem repeats within microsatellite regions” or similar
Line 54. “The first genetic loci of LS were mapped back in the 1990s’ - please provide reference. “The first genetic loci associated with LS” perhaps?
Line 54. “Nowadays, it is estimated that 1 every ̴ 300 individuals can carry of pathogenic variants in an MMR gene in North America” - editing of English language required
Line 57. “Beyond CRC and endometrial cancer, LS individuals can develop multiple primary tumors including cancers of the stomach, ovaries, urinary tract, and pancreas”. What about possible association between mutations in so called “LS predisposition genes” and hereditary breast and ovarian cancers?
Line 60. “excellent survivals”-- what does it mean?
Line 61. “revolutionizing the modern oncology” I would not say that the fact that one particular sub-type of cancer might be responsive to one type of immunotherapy revolutionizes the modern oncology.
Line 62. “Through the progressive knowledge on LS many carcinogenic mechanisms (i.e., two-hit paradigm) have been and continue to be characterized. “ - this sentence is out of context.
Note: EpCAM is the abbreviated name of the epithelial cell adhesion molecule, but EPCAM is the name of the corresponding gene.
All gene names should be italicised.
Line 75. “constitutional pathogenic mutations resulting from deficiency of MMR proteins” - mutations can not result from deficiency of proteins.
Line 87. “MSH2 is located on chromosome 2 and was first discovered in 1997” Do the authors mean that the MSH2 gene was discovered in 1997?
Line 90. “constitutional epimutations” - please provide brief explanation (define “epimutations”)
Note: please always use LS abbreviation throughout the text
Line 117 “Mutations in MLH1 and MSH2, especially those with deletions of the exon encoding for EpCAM” ??????
Table 1, Table 2. EPCAM gene is absent. Why? Breast cancer is not discussed. Why?
Line 538. “lack of high-quality evidence for extra-colonic cancer screening” - this statement is arguable
When talking about immunotherapy, every time when you mention antibody (for example, pembrolizumab), please also mention its antigen.
Line 548. “In our view, testing for dMMR/MSI-H stands out as a prime illustration of a universally applicable somatic test crucial for both therapeutic interventions and genetic counseling/testing recommendations” - nothing novel, nowadays genetic testing/counselling, including LS cases, is a part of clinical routine
Line 555. “here are promising developments in the use of immunotherapy and anti-cancer vaccines for treating these highly immunogenic tumors associated with LS" - there are only a few studies of anti-cancer vaccines as a potential therapy for LS, mostly in mice
Overall, in my opinion the text is hard to read, it requires extensive English editing, and should be re-written. The text is too repetitive. The conclusion is too broad.
Comments on the Quality of English Language
Extensive editing of English language required
Author Response
Response to the editor(s) of ‘Cancers’
Dear Editor(s),
We sincerely thank you for giving us the opportunity to consider for re-submission and potential publication a revised version of our original manuscript entitled “Lynch syndrome: from multidisciplinary management to precision prevention" by Dal Buono et al.
We kindly thank the Reviewers for the precious comments. We are pleased to know that you appreciated the topic of our manuscript. The manuscript has been significantly revised and improved according to the received suggestions. Included below you can find a point-by-point response to the remarks.
Sincerely,
Arianna Dal Buono, MD
Humanitas Research Hospital – IRCCS
Rozzano, MI, Italy
Reviewer#3
The review Lynch syndrome: from multidisciplinary management to precision prevention by Arianna Dal Buono et al is an attempt to summarize and discuss “the genotype-driven strategies for surveillance, personalized cancer prevention and treatment” in case of Lynch syndrome (LS) cancers.
Please find below my comments and suggestions
Abstract “Background and aims: Lynch syndrome (LS) is currently the most prevalent hereditary cancer condition” - what about hereditary breast and ovarian cancer syndrome? Please provide references regarding prevalence of the most common hereditary cancer conditions.
Re: Thank You for Your comment. We improved the text accordingly. Data on prevalence of LS are reported in the introduction paragraph.
“Significant advances have recently improved the understanding of the molecular carcinogenesis of LS tumors “ - what advances? The sentence is not informative.
Re: Thank You for Your comment. We improved the text accordingly.
“We aim to review the latest progresses in LS and discuss their implications for genotype-driven strategies for surveillance, personalized cancer prevention and treatment.” what implications? This sentence should be re-written too.
Re: Thank You for Your comment. We improved the text accordingly.
Line 49 - “DNA mismatch repair (dMMR)”, but “impaired DNA mismatch repair (MMR)” line 27. The abbreviation is inconsistent. Does the latter abbreviation means “deficient DNA mismatch repair (dMMR)?
Re: Thank You for Your comment. We improved the text accordingly.
Line 50. “Germline pathogenic variants <> results” - perhaps “result”?
Re: Thank You for Your comment. We improved the text accordingly.
Line 52. “microsatellite instability (MSI), otherwise modifications in the length of tandem repeats within microsatellite regions” - I suggest replacing this sentence with “microsatellite instability (MSI), i.e. modifications in the length of tandem repeats within microsatellite regions” or similar
Re: Thank You for Your comment. We improved the text accordingly.
Line 54. “The first genetic loci of LS were mapped back in the 1990s’ - please provide reference. “The first genetic loci associated with LS” perhaps?
Re: Thank You for Your comment. We improved the text accordingly.
Line 54. “Nowadays, it is estimated that 1 every ̴ 300 individuals can carry of pathogenic variants in an MMR gene in North America” - editing of English language required
Re: Thank You for Your comment. We improved the text accordingly.
Line 57. “Beyond CRC and endometrial cancer, LS individuals can develop multiple primary tumors including cancers of the stomach, ovaries, urinary tract, and pancreas”. What about possible association between mutations in so called “LS predisposition genes” and hereditary breast and ovarian cancers?
Re: Thank You for Your comment. We improved the text as suggested (see the paragraph molecular genetics and phenotypic heterogeneity).
Line 60. “excellent survivals”-- what does it mean?
Re: Thank You for Your comment. We corrected the text accordingly.
Line 61. “revolutionizing the modern oncology” I would not say that the fact that one particular sub-type of cancer might be responsive to one type of immunotherapy revolutionizes the modern oncology.
Re: Thank You for Your comment. We corrected the text accordingly.
Line 62. “Through the progressive knowledge on LS many carcinogenic mechanisms (i.e., two-hit paradigm) have been and continue to be characterized. “ - this sentence is out of context.
Re: Thank You for Your comment. We corrected the text accordingly.
Note: EpCAM is the abbreviated name of the epithelial cell adhesion molecule, but EPCAM is the name of the corresponding gene.
Re: Thank You for Your comment. We corrected the text accordingly.
All gene names should be italicised.
Re: Thank You for Your comment. We corrected the text accordingly.
Line 75. “constitutional pathogenic mutations resulting from deficiency of MMR proteins” - mutations can not result from deficiency of proteins.
Re: Thank You for Your comment. We corrected the text accordingly.
Line 87. “MSH2 is located on chromosome 2 and was first discovered in 1997” Do the authors mean that the MSH2 gene was discovered in 1997?
Re: Thank You for Your comment. MSH2 was mapped and cloned in 1993 (see Fishel R, Lescoe MK, Rao MR, Copeland NG, Jenkins NA, Garber J, Kane M, Kolodner R. The human mutator gene homolog MSH2 and its association with hereditary nonpolyposis colon cancer. Cell. 1993 Dec 3;75(5):1027-38).
Line 90. “constitutional epimutations” - please provide brief explanation (define “epimutations”)
Re: Thank You for Your comment. We improved the text accordingly.
Note: please always use LS abbreviation throughout the text
Re: Thank You for Your comment. We improved the text accordingly.
Line 117 “Mutations in MLH1 and MSH2, especially those with deletions of the exon encoding for EpCAM” ??????
Re: Thank You for Your comment. We corrected the text accordingly.
Table 1, Table 2. EPCAM gene is absent. Why? Breast cancer is not discussed. Why?
Re: Thank You for Your comment. Breast cancer is almost never included on the LS increased cancer risks table (see NCCN guidelines). There are insufficient data supporting an increased risk for breast cancer for women with Lynch syndrome (Engel C, et al. J Clin Oncol 2012;30:4409-4415; Barrow E, et al. Clin Genet 2009;75:141-149; Dominguez-Valentin M, et al. Genet Med 2020;22:15-25; Harkness EF, et al. J Med Genet 2015;52:553-556; Hu C, et al. N Engl J Med 2021;384:440-451; Dorling L, et al. N Engl J Med 2021;384:428-439; Stoll J, et al. J Clin Oncol 2020;4:51-60). As a result, breast cancer is not included on the LS increased cancer risks table. Breast cancer risk management should be based on personal and family history.
Line 538. “lack of high-quality evidence for extra-colonic cancer screening” - this statement is arguable
Re: Thank You for Your comment, we corrected the text accordingly
When talking about immunotherapy, every time when you mention antibody (for example, pembrolizumab), please also mention its antigen.
Re: Thank You for Your comment, we corrected the text accordingly
Line 548. “In our view, testing for dMMR/MSI-H stands out as a prime illustration of a universally applicable somatic test crucial for both therapeutic interventions and genetic counseling/testing recommendations” - nothing novel, nowadays genetic testing/counselling, including LS cases, is a part of clinical routine
Re: Thank You for Your comment. This is in the conclusion section and can remain unchanged.
Line 555. “here are promising developments in the use of immunotherapy and anti-cancer vaccines for treating these highly immunogenic tumors associated with LS" - there are only a few studies of anti-cancer vaccines as a potential therapy for LS, mostly in mice
Re: Thank You for Your comment, we corrected the text accordingly
Overall, in my opinion the text is hard to read, it requires extensive English editing, and should be re-written. The text is too repetitive. The conclusion is too broad.
Re: Thank You for Your comment, we had the text reviewed and corrected by a colleague who is a native-speaking.
Editor
(I) Please check that all references are relevant to the contents of the manuscript.
Re: Thank You for Your comment, we checked all references.
(II) Any revisions to the manuscript should be highlighted, such that any changes can be easily reviewed by editors and reviewers.
Re: Thank You for Your comment, we highlighted all revisions.
(III) Please provide a cover letter to explain, point by point, the details of the revisions to the manuscript and your responses to the referees’ comments.
Re: Thank You for Your comment, we provided a cover letter with a point by point response.
(IV) If you found it impossible to address certain comments in the review reports, please include an explanation in your appeal.
Re: Thank You for Your comment, all the comments of the reviewers have been extensively addressed.
(V) The revised version will be sent to the editors and reviewers
Re: Thank You for Your comment, we will send the revision to whom it may concern.
Round 2
Reviewer 2 Report
Comments and Suggestions for Authors
the authors have sufficiently addressed the reviewers' points. No additional comments at this time.
Reviewer 3 Report
Comments and Suggestions for Authors
The authors addressed all my previous comments and suggestions, the manuscript can be published in its current form